

# Spectral Induced Polarization survey for the estimation of hydrogeological parameters in an active rock glacier

Clemens Moser[1], Umberto Morra di Cella[2], Christian Hauck[3], and Adrián Flores Orozco[1]

[1]Research Unit Geophysics, Department of Geodesy and Geoinformation, TU Wien, Vienna, Austria
[2]Climate Change Unit, ARPA VdA (Environmental Protection Agency of Valle d'Aosta), Aosta valley, Italy
[3]Department of Geosciences, University of Fribourg, Fribourg, Switzerland

*Correspondence to*: Clemens Moser (clemens.moser@geo.tuwien.ac.at)

**Abstract.** Degrading permafrost in rock glaciers has been reported from several sites in the European Alps. Changes in ground temperature and ice content are expected to affect the hydrogeological properties of the rock glacier and in turn modify the runoff regime and groundwater recharge in high-mountain environments. In this study, we investigate the use of an emerging geophysical method to understand the hydrogeological properties of the active Gran Sometta rock glacier, which consists of a two lobe-tongue (a white and a black) differing in their geologies. We present the application of the spectral induced polarization (SIP) imaging, a method that provides continuous spatial information about the electrical conductivity and polarization of the subsurface, which are linked to hydrogeological properties. To quantify the water content and the hydraulic conductivity from SIP imaging results, we used the petrophysical dynamic stern layer model. The SIP results show a continuously frozen layer at 4-6 m depth along both lobes which hinders the infiltration of water leading to a quick flow through the active layer. To evaluate our results, we conducted tracer experiments monitored with a time-lapse electrical conductivity imaging which confirms the hydraulic barrier associated with the frozen layer and allows to quantify the pore water velocity (~$10^{-2}$ m/s). Below the frozen layer, both lobes have distinct water content and hydraulic conductivity. We observed a higher water content in the black lobe, which moves faster than the white lobe supporting the hypothesis that the water content at the shear horizon dominates rock glacier velocity. Our study demonstrates that the SIP method is able to provide valuable information for the hydrogeological characterization of rock glaciers.

## 1 Introduction

The European Alps are undergoing drastic changes due to climate change (Beniston et al., 2018) such as retreating glaciers, the degradation of permafrost or the loss of ground ice (e.g., Biskaborn et al., 2019). The loss of ground ice has been linked to rockfalls and slope instability (e.g., Krautblatter et al., 2013; Haeberli et al., 2017); thus, poses a threat to infrastructure and human life not only in high-mountain environments but also for downstream areas. Active rock glaciers, which consist of frozen rocks, sediments and large amounts of ground ice, move downslope resulting in surface deformation (e.g., Haeberli, 1985; Barsch, 1996). Accordingly, they can play an important role in the storage of frozen and liquid water (Jones et al., 2018; Wagner et al., 2021). Rock glaciers are more resilient towards the increase of air temperatures, compared



to glaciers, as they are protected by a top layer of large blocks and air-filled voids with low thermal conductivity (Haeberli et al., 2006; Giardino et al., 2011; Amschwand et al., 2023). Due to the higher resilience of subsurface ice towards increasing air temperatures compared to glacier ice, future predictions expect a change in the hydrological regime in mountainous areas with a shift from glacially dominated hydrological systems to periglacially dominated systems (e.g., Haeberli et al., 2017),
which has a particular impact on areas which are supplied by water from high-mountain environments.

Water, coming from snow meltwater, rainfall, groundwater, and ice meltwater, commonly percolates through intact rock glaciers as supra-permafrost flow in the active layer along the top of an ice-rich core, which is assumed to be quasi-impervious (e.g., Giardino et al., 2011; Buchli et al., 2013; Winkler et al., 2016; Krainer et al., 2007; Jones et al., 2019). The supra-permafrost flow is also called quickflow because coarse-grained materials in the active layer are characterized by high
hydraulic conductivity, with reported values in pore water velocity ranging between $10^{-3}$ and $10^{-2}$ m/s (e.g., Krainer and Mostler, 2002; Buchli et al., 2013; Winkler et al., 2016; Harrington et al., 2018; Del Siro et al., 2023). Due to their high hydraulic conductivities, rock glaciers release water from heavy rainfalls quickly and they increase flood peaks when compared to catchments without rock glaciers (Geiger et al., 2014; Pourrier et al., 2014). Within rock glaciers, the ice-rich cores might be considered as quasi-impermeable as they hinder the infiltration of water from rainfalls or snowmelt. In
contrast, groundwater can move underneath the frozen layer as sub-permafrost flow (also called baseflow) through fine-grained sediments (e.g., Arenson et al., 2002; Krainer et al., 2015) with lower hydraulic conductivity than the supra-permafrost layer (Winkler et al., 2016; Rogger et al., 2017). In degraded rock glaciers with only discontinuous ice layers, also rainwater and snow meltwater can percolate into deeper areas, where it accumulates leading to an increased storage capacity in the fine-grained section of the rock glacier (Pourrier et al., 2014; Winkler et al., 2016).

The monitoring of rock glacier kinematics during the last decades in the European Alps revealed an overall acceleration in the downslope movement (e.g., Delaloye et al., 2010; Wirz et al., 2014; Kellerer-Pirklbauer et al., 2024), which was first related to warming ice in rock glaciers due to increasing air temperatures (Kääb et al., 2007; Roer et al., 2008). However, recent studies demonstrate that rock glacier movement is rather dominated by the liquid water content around the shear horizon (Wirz et al., 2016; Cicoira et al., 2019). This is because most of the deformation (60-90%) occurs in the shear
horizon underneath the ice-rich layer (Arenson et al., 2002; Cicoira et al., 2020), where groundwater can accumulate (Kenner et al., 2017; Buchli et al., 2018). Water accumulated within the rock glacier increases the pore water pressure and decreases the frictional resistance leading to deformation (Cicoira et al., 2019; Kenner et al., 2017). Ikeda et al. (2008) and Cicoira et al. (2020) demonstrated that in degrading rock glaciers water flow paths arise in thawed areas. Water from precipitation and snow melt can slowly infiltrate through such flow paths down to the shear horizon on the bottom of the rock glacier.

To-date, the internal structure of rock glaciers is investigated either by boreholes or geophysical measurements. While boreholes can provide direct information about the subsurface, e.g., in terms of ice and water content, they only provide point-wise information. Hence, borehole data need to be interpolated to resolve spatial variations in the internal structure, which may bias the interpretation of the data, as those parameters can vary strongly in rock glaciers (e.g., Krainer et al.,





2015). Moreover, core drillings in rock glaciers are costly and very rare (e.g., Noetzli et al., 2021), and drilling in a given
position is possible only once; thus, not suited for monitoring applications.

To overcome the point-wise information derived from temperature sensors in boreholes, Phillips et al. (2023) installed a
cross-borehole electrical resistivity tomography (ERT) monitoring on an active rock glacier. ERT is a geophysical method,
which provides continuous 2D or 3D spatial information about the subsurface electrical resistivity or its inverse, the
electrical conductivity; thus, it is a common tool for the detection of frozen areas because of the low electrical conductivity
of ice compared to non-frozen areas (e.g., Kneisel et al., 2008; Mollaret et al., 2019). Liquid water content and electrical
conductivity are directly linked as the electrical conductivity increases with increasing liquid water saturation (Archie,
1942). To quantify ice, but also air and water content, Hauck et al. (2011) proposed the so-called 4-phase model, which
combines the electrical conductivity data with seismic information about the velocity of P-waves. Apart from several
applications for ground ice investigations, Halla et al. (2021) used the 4-phase model to identify water pathways in an active
rock glacier in the Andes and related them to surface deformation and kinematics of the rock glacier. The 4-phase model was
further integrated in a joint inversion framework to estimate ice, water, air content and porosity simultaneously (Wagner et
al., 2019; Mollaret et al., 2020; Steiner et al., 2021; Pavoni et al., 2023a). However, in its original formulation the model
neglects variations in the surface conductivity along the interface between grains and pore water (Waxman and Smits, 1968).
When neglecting surface conductivity, the 4-phase model may overestimate the water content, as a decrease in the electrical
resistivity would only be associated to an increase of interconnected pore water. Mollaret et al. (2020) applied the joint
inversion framework on several different sites using different formulations of the electrical response of the four phases, one
of them considered surface conductivity but only through a constant term.

The surface conductivity at low frequencies (<1 kHz) in the subsurface is due to the presence of the electrical double layer
(EDL) at the grain-electrolyte interface (e.g., Waxman and Smits, 1968). During the application of an electrical field, the
EDL contributes to the accumulation and polarization of charges and the development of salinity gradients that contribute to
the electrical conduction. While the electrical conductivity derived from ERT consists of the electrolytic and surface
conductivity, the polarization of the subsurface is only related to the surface area of particles, i.e., to the surface conductivity
(Vinegar and Waxman, 1984; Revil and Florsch, 2010). As an extension of the ERT method, the induced polarization (IP)
method permits to measure both the conductive and capacitive properties of the subsurface at low frequencies (< 10 kHz),
either in the frequency- (FDIP) or time-domain (TDIP) (for a review of the method see Binley and Slater, 2020). Different
materials with different textural properties polarize at different length-scales, which reflects in the frequency dependency of
the electrical properties (see Binley et al., 2005; Revil and Florsch, 2010). For the analysis of the frequency dependence,
complex conductivity measurements over a broad range of frequencies (mHz – kHz range) are necessary within the so-called
spectral IP (SIP) method.

Grimm and Stillman (2015) demonstrated for the first time that field-scale resistivity measurements at different frequencies
can be used to delineate areas with high ice content on the field scale. They evidenced that the so-called frequency effect
(FE) in resistivity measurements, which is the relative difference between the resistivity at a high and a low frequency,





increases with decreasing temperature as well as with ice content. The FE is a clear indication of induced polarization phenomena and has been used for decades in mineral exploration to delineate areas with high metal content (e.g., Seigel et

al., 1959). Grimm and Stillman (2015) measured a FE of up to 0.9 between resistivity measured at 10 Hz and 19 kHz in areas with an ice content around 90% due to the polarization of ice at high frequencies (Auty and Cole, 1952; Stillman et al., 2010; Coperey et al., 2018; Limbrock et al., 2020). However, the authors did not investigate the SIP response due to unfrozen rocks and rocks with different ice content. Maierhofer et al. (2022) directly measured the polarization effect at a frequency of 75 Hz to improve the discrimination between frozen and unfrozen ground in an ice-bearing talus slope in

Switzerland. Mudler et al. (2022) investigated the application of IP measurements at high frequencies (kHz range) to quantify ice content directly based on the relaxation frequency of ice (Zorin and Ageev, 2017). However, to the best of our knowledge, no study examined the potential of IP data for the estimation of water content and hydraulic conductivity in a rock glacier or other alpine permafrost landforms yet.

Hydrogeological information of rock glaciers is critical to fully understand the parameters controlling their deformation and

movement. Hence, the objective of this study is to understand the hydrogeological properties of the active rock glacier Gran Sometta in the Italian Alps. We evaluate the potential of SIP imaging results for the estimation of hydrogeological parameters, namely the hydraulic conductivity and the water content. The SIP results are validated by two tracer experiments, where the distribution and flow of a saltwater injection was monitored by repeated 3D ERT images permitting the direct estimation of the pore water velocity in the rock glacier. Additionally, we investigate the link between the hydro-

geophysical results and spatial variations in deformation rates of the Gran Sometta rock glacier to demonstrate the relevance of geophysical investigations to understand the main drivers of rock glacier movement in the European Alps.

In pharetra massa dictum gravida scelerisque. Sed vitae purus eget purus tincidunt accumsan ut at magna.

## 2 Material and Methods

### 2.1 Study area

The Gran Sometta rock glacier is a tongue-shaped rock glacier located in the Valtournenche valley (Aosta valley) in the Western European Alps (see Fig. 1) at an elevation between 2630 and 2770 m. The rock glacier has a total length of 400 m, a width of 150-300 m and a thickness of 20-30 m, estimated from the height of the front (Bearzot et al., 2022). At the surface, the rock glacier mainly consists of pebbles and angular blocks originating from the rock walls of the Gran Sometta peak. In the central area, longitudinal ridges dominate the structure of the morphology, while in the front part there are transversal

furrows. The upper (southern) part of the rock glacier was covered by a glacier during the Little Ice Age (LIA), which still has effects on the distribution of ice content (see detailed descriptions of the rock glacier in Bearzot et al., 2022 and Bearzot et al., 2023).

The rock glacier can be split into two main lobes (see Fig. 1), which are called the white lobe (eastern part) and the black lobe (western part) as the former is mainly composed of light-colored dolomitic marbles and the latter of dark-colored





carbonate-silicate schists. Apart from the color, the lobes differ in their internal structure and their kinematics (Bearzot et al.,
2022). While in the black lobe an ERT survey revealed a continuous ice-rich layer with a thickness of 20 m approximately,
in the white lobe there are more spatial variations in the ice content, possibly related to the influence of the coverage of a
LIA glacier. The rock masses in these areas possibly warmed up during the LIA due to the thermal isolation of the ice cover
(Bearzot et al., 2022). Kinematic investigations of Bearzot et al. (2022) for the period 2013-2020 show that the black lobe

moves faster (~1 m/y) than the white lobe (~0.5 m/y), which has been linked to the different internal structure and the steeper
topography of the black lobe.

**Figure 1: Map of the Gran Sometta rock glacier, which consists of two lobes, the black lobe in the west and the white**

**lobe in the east. The SIP profiles P1-P8 are indicated by red lines, while profiles of the small-scale tracer experiments**
**are indicated by green lines. The large-scale tracer profile is presented by a yellow line and the positions of two**
**boreholes (BH1 and BH2) are displayed by white dots. Lower right is a map of the Alps with the position of the Gran**
**Sometta rock glacier. A 3D visualization of the area of the tracer experiment on the white lobe (TW) is given in the**





## 2.2 The spectral induced polarization method

IP measurements are an extension of the resistivity method and provide the conductive and capacitive properties of the subsurface (for an overview see Binley and Slater, 2020). When working in the frequency-domain, as has been done in this study, an alternating current with a certain angular frequency $\omega$ is injected via two electrodes and the resulting voltage is

measured between two additional electrodes. The ratio between the voltage and the injected current gives a complex-valued transfer impedance $Z^*(\omega)$ consisting of a magnitude $|Z^*(\omega)|$ and a phase shift $\varphi(\omega)$ between the voltage and the current.

$$Z^*(\omega) = |Z^*(\omega)|e^{i\varphi(\omega)} \tag{1}$$

When using the SIP method, measurements are not only conducted at one frequency, but in a broad range of frequencies, typically in the range between 1 mHz and 10 kHz to provide additional information about the frequency dependence of the

electrical properties. Inversion of measurements from hundreds or thousands of electrode combinations, also called quadrupoles, gives a distribution of the complex-valued electrical conductivity $\sigma^*$ or its inverse, the complex resistivity $\rho^*$. The complex conductivity is used to represent both electrical properties of the subsurface at the low frequency: the energy conduction (presented in the real component $\sigma'$) and the energy stored due to the polarization of charges accumulated at the EDL (imaginary component $\sigma''$). The $\sigma^*$ can also be represented by its magnitude $|\sigma^*|$ and phase angle $\varphi(\omega)$:

$$\sigma^*(\omega) = \sigma'(\omega) + i\sigma''(\omega) = |\sigma^*(\omega)|e^{i\varphi(\omega)} \tag{2}$$

In areas free of metallic materials $|\sigma^*|$ is approximately equal to $\sigma'$ because $\varphi(\omega)$ is relatively low (typically below 100 mrad) (Binley and Slater, 2020). Current can be conducted through the subsurface by three different mechanisms (see Eq. (3)): (1) The matrix conductivity $\sigma_m$ describes current conduction through the solid (i.e., grains and matrix). This mechanism is negligible at sites without electronic conductors and semi-conductors like metallic minerals, such as in rock glaciers. (2)

The electrolytic conductivity $\sigma_{el}$ (Eq. (4)) describes current conduction through the pore fluid, which depends on the porosity $\phi$, the cementation exponent $m$ (which can be also expressed by the formation factor $F = \phi^{-m}$), the saturation $S_w$, the saturation exponent $n$ and the fluid conductivity $\sigma_f$ (e.g., Archie, 1942). (3) The surface conductivity $\sigma_s^*$ describes the conduction, represented by the real part $\sigma_s'$, and the polarization, represented by the imaginary part $\sigma_s''$, in the electric double layer (EDL) at the interface between solid particles and the pore fluid. $\sigma_s^*$ is dominated by the surface area and surface

charge of particles and is frequency dependent (e.g., Waxman and Smits, 1968).

$$\sigma'(\omega) = \sigma_m + \sigma_{el} + \sigma_s'(\omega) \tag{3}$$

$$\sigma_{el} = S_w^n F^{-1}\sigma_f \tag{4}$$

The imaginary component of the complex conductivity is only dependent on the imaginary component of the surface conductivity because the pore water is assumed to be non-polarizable at frequencies below 10 kHz:

$$\sigma''(\omega) = \sigma_s''(\omega) \tag{5}$$



In the absence of electronically conductive materials, the polarization is only related to the surface area and the surface charge of the particles (Vinegar and Waxman, 1984; Leroy et al., 2008; Revil and Florsch, 2010).

### 2.3 Deriving hydraulic conductivity from SIP measurements

The IP method provides information about textural properties of the subsurface, which control the ability of water to flow,
i.e., its hydraulic properties (see Binley and Slater; 2020). According to the Kozeny-Carman equation, geometric parameters, i.e., the hydraulic radius and the tortuosity, are dominating the hydraulic conductivity of the subsurface (Carman, 1939; Kozeny, 1927). These parameters can be derived from IP and SIP measurements as the tortuosity is linked to the electrolytic conductivity and the hydraulic radius is related to the surface conductivity (for a review see Slater, 2007). Using only electrical conductivity can lead to an ambiguity as in the case of relatively coarse-grained materials, e.g., sands and gravels,
the electrolytic conductivity dominates over the surface conductivity resulting in a positive correlation between electrical and hydraulic conductivity (e.g., Frohlich et al., 1996). If the materials are relatively fine-grained (clays and fine silts), the surface conductivity dominates over the electrolytic conductivity leading to an opposite correlation because the pores are less connected hindering water flow (e.g., Urish et al., 1981). Accounting for surface conductivity through the use of IP measurements improves hydraulic conductivity estimations, as demonstrated in the laboratory (e.g., Binley et al., 2005;
Revil and Florsch, 2010; Slater et al., 2014; Weller et al., 2015) and on the field scale (e.g., Hördt et al. 2009; Benoit et al., 2019; Soueid Ahmed et al., 2020; Kemna et al., 2004).

Several studies have demonstrated an improvement of the hydraulic conductivity estimation when considering the frequency dependence of the polarization, because the frequency dependence is particularly linked to different length scales of the pore space (e.g., Binley et al., 2005; Revil et al., 2015). Thus, SIP measurements permit an improved estimation of the hydraulic
properties, as observed from laboratory (e.g., Revil and Florsch, 2010; Slater et al., 2014; Weller et al., 2015) and field scale investigations (e.g., Hördt et al. 2009; Benoit et al., 2019; Flores Orozco et al., 2022). However, investigations in different environments demonstrate that there is no universal model linking SIP parameters and hydraulic conductivity, thus, every model applied on the field needs to be carefully evaluated (see Slater, 2007).

### 2.4 The relation between complex conductivity and hydrogeological parameters

The volumetric water content $\theta$ of porous media is directly linked to the electrical conductivity $\sigma'$ via $F$, $S_w$ and $n$ (see Eq. (4)), which allows to estimate $\theta$ based on $\sigma'$. However, as described above, $\sigma'$ is not only related to $\sigma_{el}$ but also to $\sigma'_s$ (see Eq. (3)). To take $\sigma'_s$ into account, in this study we use the dynamic stern layer model DSLM (Revil, 2013a,b), which provides the following equations for the instantaneous (high frequency) conductivity $\sigma_\infty$ and the direct current (low frequency) conductivity $\sigma_0$, where the first term represents the frequency independent $\sigma_{el}$ and the second term the frequency dependent
$\sigma'_s$:

$$\sigma_\infty = \theta^m \sigma_f + \theta^{m-1} \rho_g B \, CEC \tag{6}$$





$$\sigma_0 = \theta^m \sigma_f + \theta^{m-1} \rho_g (B - \lambda) \, CEC \tag{7}$$

where $\rho_g$ is the grain density (kg m$^{-3}$), $B$ is the apparent mobility of the counterions for surface conduction (m$^2$ s$^{-1}$ V$^{-1}$) and $\lambda$ is the apparent mobility of the counterions for the polarization (m$^2$ s$^{-1}$ V$^{-1}$). $CEC$ is the cation exchange capacity (C kg$^{-1}$),

which is directly linked with the clay content (Mao et al., 2016) and the normalized chargeability $M_n$, which is the difference between $\sigma_\infty$ and $\sigma_0$, but can also be calculated by the difference between the conductivity at a high frequency $f_2$ and the conductivity at a low frequency $f_1$.

$$M_n = \sigma_\infty - \sigma_0 = \theta^{m-1} \rho_g \lambda \, CEC \tag{8}$$

In the case of a broad distribution of grain sizes, the spectrum of the conductivity phase angle is rather flat and can be

roughly described by a constant phase angle model, the so-called Drake's model (Van Voorhis et al., 1973). If such model is applicable, $M_n$ is linearly related to $\sigma''$ (at the geometric mean frequency between $f_1$ and $f_2$) by the factor $\alpha$, as demonstrated by laboratory and field investigations (see Revil et al., 2017; Revil et al., 2021)

$$\sigma'' = -\frac{M_n}{\alpha} \tag{9}$$

$$\alpha = \frac{2}{\pi} \ln(A), \tag{10}$$

where $A$ is the number of decades between $f_1$ and $f_2$.

If we combine equations 6 and 8, we can formulate expressions for $\theta$ and $CEC$ (Revil et al., 2020):

$$\theta = \left[ \frac{1}{\sigma_f} \left( \sigma_\infty - \frac{M_n}{R} \right) \right]^{1/m} \tag{11}$$

$$CEC = \frac{M_n}{\theta^{m-1} \rho_g \lambda} \tag{12}$$

where $R$ is the ratio between $\lambda$ and $B$. Laboratory investigations have shown that $R$ is a dimensionless constant value in a

range of 0.1±0.02 (Revil et al., 2017a,b,c). To determine $\sigma_f$ for equation 11, we measured $\sigma_f$ at the spring of the white and black lobe, which is in a range of 0.02-0.03 S/m. To take into account the increase in the salinity of the water stored in the rock glacier in contact with rock material, we used a value of 0.01 S/m, as the measured fluid conductivities represent the value measured at the rock glacier spring at its maximum. For $m$ we used 2, as used by Coperey et al. (2019), and for $\rho_g$ we used the mean grain density of green schist, dolomite and marble (2700 kg m$^{-3}$).

Soueid Ahmed et al. (2020) further developed the DSLM for the estimation of the permeability $k$ (m$^2$) in unsaturated media based on $\theta$, $CEC$ and the fitting parameter $k_0$.

$$k \approx \frac{k_0 \theta^6}{\left( \rho_g CEC \right)^2} \tag{13}$$

$\theta$ and $CEC$ were calculated by equations 11 and 12, for $k_0$ we used a value of $10^{4.3}$ as estimated in the laboratory by Soueid Ahmed et al. (2020). In a next step, we converted $k$ into the hydraulic conductivity $K$ (m/s) by the following equation:

$$K = \frac{g\delta}{\mu} k \tag{14}$$



where we used reference values for the gravitational acceleration $g$ ($g = 9.81$ m s$^{-2}$) and for the groundwater dynamic viscosity $\mu$ ($\mu = 1.0 \times 10^{-3}$ kg m$^{-1}$ s$^{-1}$). For the density of the fluid $\delta$, we assumed 1000 kg m$^{-3}$. $K$ can be converted into the pore water velocity $v_p$ under the consideration of the porosity $\phi$, where we used a mean value of 40% (as used as an approximate value in other rock glacier studies, e.g., Hauck et al., 2011; Halla et al., 2021) and the hydraulic gradient $\frac{\Delta H}{\Delta L}$.

The hydraulic gradient is the ratio between the difference in height $\Delta H$ and the horizontal distance $\Delta L$ between two points and which we estimated from a digital elevation model:

$$v_p = \frac{K}{\phi}\frac{\Delta H}{\Delta L} \qquad (15)$$

## 2.5 SIP field measurements and data processing

In October 2022, we collected SIP data along eight lines (P1 to P8) on the Gran Sometta rock glacier. P5, P6 and P8 are
longitudinal lines and P1, P2, P3, P4 and P7 are transversal to the flow direction of the rock glacier (see Fig. 1). Different line directions allow to collect data of quadrupoles with different dipole orientations, which provides a good coverage of the subsurface electrical properties in all directions (see e.g., Chambers et al., 2002; Moser et al., 2023). For each line of electrodes, we used 32 stainless-steel electrodes with a separation of 7.5 m. We used two configurations: a dipole-dipole (DD) normal and reciprocal (current and voltage dipole changed) configuration (see e.g., LaBrecque et al., 1996) with a
dipole length of four times the electrode spacing, and a multiple-gradient (MG) configuration. With the DD configuration, we collected data in a frequency range of 0.5-225 Hz, with the MG configuration we used a frequency range of 0.1-225 Hz. For the data acquisition, we used the eight-channel device DAS-1 (Data Acquisition System, from MPT-IRIS Inc.), which was connected to stainless steel electrodes via coaxial cables (COAX10, designed at TU Wien; see Flores Orozco et al., 2021). The isolation of coaxial cables permits to avoid electromagnetic (EM) coupling due to cross-talk between the cables
as demonstrated by Flores Orozco et al. (2013 and 2021), and (in alpine permafrost) by Maierhofer et al. (2022).

The SIP data were processed in four steps: (1) We calculated the geometric factors $k_{num}$ as the ratio between numerically modeled transfer resistances of a homogeneous synthetic model and the resistivity of the homogeneous model. Readings with an inconsistent polarity between $k_{num}$ and the measured transfer resistances were deleted as erroneous readings. (2) All readings with an absolute misfit between normal and reciprocal above two times the standard deviation of the normal-
reciprocal misfit for the entire data set, as well as those readings with a relative normal-reciprocal misfit (above 5% in the case of $|Z^*|$, and 50% in the case of $\varphi$) were removed, similar to the filtering protocol of Flores Orozco et al. (2019). (3) Additionally, outliers in the readings of apparent resistivity $|\rho_a^*|$, defined as those below $10^3$ $\Omega$m and above $10^5$ $\Omega$m, as well as positive impedance phase values ($\varphi$) were removed. Steps 1 to 3 were conducted separately for each frequency. (4) Equal to the routine of Moser et al. (2023), readings, which were removed during steps 1 to 3 in at least one frequency, were
removed in all frequencies to keep the same quadrupoles for the inversion of data collected at all frequencies, aiming at having consistent sensitivity in the inverted images (Flores Orozco et al., 2013). After applying filtering steps (1) – (4) 731 of 1792 quadrupoles remained in the case of the MG configuration and 359 of 4416 in the case of the DD configuration.



Considering the high density of measurements collected with the MG configuration designed in our study, the number of filtered measurements has just a minimal effect on the coverage and resolution of our data sets, as demonstrated, for
instance, in the pseudosection (in terms of $|\rho_a^*|$) presented in the Appendix (see Fig. A1). From now on, for all multi-frequency analyses, we only consider MG data because of the larger range of frequencies used for the data collection with the MG configuration (0.1-225 Hz) than with the DD configuration (0.5-225 Hz).

We inverted the data collected at each frequency independently using the ResIPy code (Blanchy et al., 2020), which calls the cR3t 3D complex resistivity inversion algorithm (for details see Binley and Slater, 2020). cR3t is a smoothness-constrained
algorithm based on complex calculus, which iteratively solves for a 3D complex conductivity model. The 3D model is represented by a finite element mesh, which was created in Gmsh (Geuzaine and Remacle, 2009). The mesh is based on tetrahedral elements, whose sizes increase with increasing distance from the electrodes, and it incorporates the topography by a digital terrain model. The inversion algorithm fits the data under the consideration of an error model to a certain level of confidence (Binley and Kemna, 2005), which is here represented by the error-weighted root-mean square error (RMS)
between the data and the forward solution of the inverse model. The error model consists of a relative magnitude error, which for our measurements was estimated at 10%, as well as an absolute phase error, which in our measurements was estimated at 7 mrad (at 0.1 Hz) and increased to 43 mrad (at 25 Hz) due to the increasing contamination of EM coupling with increasing frequency in the data (e.g., Binley et al., 2005). All inversions converged and resulted in an error-weighted RMS value equal or close to 1.

**2.6 Resistivity tracer experiment**

Several studies have demonstrated the value of monitoring saltwater tracer tests by time-lapse ERT to investigate hydraulic connections between different geological units or to directly estimate hydraulic conductivity of the subsurface (e.g., Kemna et al., 2002; Singha and Gorelick, 2005; Cassiani et al., 2006; Perri et al., 2012) because the electrical conductivity is sensitive to saltwater due to its high fluid conductivity. Commonly, the dispersion of a tracer is monitored through chemical
analyses of samples from several sampling points, e.g., boreholes. However, such information gives only discrete information about the tracer flow and can fail if the tracer escapes from the sampling points (Cassiani et al., 2006). Coupling tracer experiments with time-lapse ERT monitoring overcomes such limitation as the method provides continuous images of the subsurface over time, which can be used to describe the movement and the dispersion of the fluid (e.g., Camporese et al., 2011).

We carried out three water amendments into the subsurface, two small and one large tracer tests, identified in such way by the volume of water injected, and the geometry of the ERT arrays for monitoring. One of the small tracer tests was conducted in the white lobe (TW) in October 2022 and the second one in the black lobe (TB) in August 2023 (see location in Fig. 1). The small tracer tests aimed at investigating the dispersion properties of water in the rock glacier, particularly to estimate the hydraulic conductivity in the active layer. For each small tracer experiment, we injected 70 l of saltwater
(29 g NaCl per liter; $\sigma_f = 41.5$ mS/cm at the white lobe and $\sigma_f = 51.5$ mS/cm at the black lobe) in 4.5 minutes and



conducted resistivity measurements every 2 minutes and 10 seconds along three lines for 3 hours and 20 minutes (see Fig. 1) in the frequency-domain at 7 Hz. We decided upon 7 Hz instead of a lower frequency to decrease the measurement time; thus, increasing the temporal resolution of the ERT monitoring. Test measurements at 7 Hz (data not shown) showed similar data quality compared to lower frequencies (1 Hz). The ERT setup consists of two parallel lines per lobe (profile TW-P1 and TW-P2, and TB-P1 and TB-P2 respectively), which are longitudinal to the rock glacier flow direction and include 16 electrodes per line, and one transversal profile per lobe (TW-P3 and TB-P3) consisting of 20 electrodes, which crosses TW-P1 and TW-P2, and TB-P1 and TB-P2 respectively, 4 m away from the injection point (see Fig. 1). For all time steps, we used a DD skip-1 protocol with inline (current and voltage dipole in one line) and crossline (current and voltage dipole in two different lines) quadrupoles along TW-P1 and TW-P2, and TB-P1 and TB-P2 respectively, and DD skip-1 inline measurements along TW-P3 and TB-P3 respectively. The number of skips is the number of electrodes skipped within a dipole. One full SIP measurement (0.5-225 Hz) was conducted before the start of the tracer injection and the 7 Hz data were used as the baseline dataset (time step 0) for the ERT monitoring. After 24 minutes, we repeated the tracer experiment with the same protocol to validate the results of injection 1.

The large tracer experiment aimed at investigating larger paths for water conduction and was conducted in August 2023. To this end, we injected 425 l saltwater ($\sigma_f$ = 51.5 mS/m) over 6 minutes on the black lobe (see Fig. 1). We repeated ERT measurements every 15 minutes along a 355 m long 2D profile (TB-P4 in Fig. 1) with an electrode separation of 5 m. ERT measurements were carried out with a DD schedule in the time-domain with a pulse length of 250 ms.

In the processing of the ERT data collected during the tracer tests, we removed measurements with a wrong polarity in all time steps and kept only those quadrupoles, which were found in all time steps to invert for imaging results with similar sensitivity, analogously to the step followed for the comparison of inversion results for different frequencies in the SIP data. For the small-scale tracer experiments, we inverted the time-lapse data of the two parallel electrode lines (TW-P1 and TW-P2; TB-P1 and TB-P2 respectively) three-dimensionally in ResIPy (Blanchy et al., 2020), with a time-lapse background constrained inversion (TLBC) under the consideration of a relative error of 10%. We used the TLBC approach because the comparison with results of a time-lapse difference approach (TLD) and the independent (IDP) inversion of all time steps showed that in areas where we expected no changes over time, we revealed less changes in the resistivity images of the TLBC approach than in the images of the IDP and TLD inversion (data not shown here). Data collected along the profiles perpendicular to the parallel lines (TW-P3 in the white lobe and TB-P3 in the black lobe) were not considered in the inversion as we did not observe any saltwater movement perpendicular to the elevation gradient (data not shown here). For the large-scale tracer experiment, we also decided upon the TLBC inversion approach as we observed smoother changes in the conductivity over time than with the IDP and TLD approach.

## 2.7 Kinematic analysis to evaluate surface deformation

The surface of the Gran Sometta rock glacier is regularly monitored by UAV surveys. They are carried out every year at the end of August to produce an orthomosaic and a digital surface model (DSM) of the rock glacier, which is used to understand





surface deformation (results from the period 2016-2019 were published in Bearzot et al., 2022). Technical details about the
UAV campaigns can be found in Table 1. The images collected are processed using a Structure-from-Motion (SfM)
workflow implemented in the commercial software Agisoft Metashape. A network of 23 painted targets was used as ground
control points (12) and check points (11). The RMSE value for each survey is shown in Table 1 (XY error). The horizontal
displacements were calculated for the periods 2020-2023 and 2022-2023 using the positions of blocks detected through
photointerpretation on orthomosaics with a resolution of 2 cm/pixel. The spatial distribution of the rock features was defined
using a square grid of 20 x 20 m. The position of the most representative targets for each year was defined manually and the
horizontal displacement of the chosen points was calculated using the Qgis plugin "PointsToPaths".

| Date | Covered area (km²) | Ground sampling distance (cm/px) | UAV model | Camera model | Number of images | Mean flying altitude (m AGL) | XY error (cm) - check points RMSE |
|---|---|---|---|---|---|---|---|
| August 21st 2020 | 0.337 | 2.33 | DJI Phantom 4 Pro V2 | FC6310 (lens 35 mm) | 665 | 85 | 1.82 |
| August 20th 2022 | 0.36 | 1.35 | DJI M300 RTK | Zenmuse DJI P1 (lens 35 mm) | 763 | 115 | 3.20 |
| August 21st 2023 | 0.36 | 1.39 | DJI M300 RTK | Zenmuse DJI P1 (lens 35 mm) | 790 | 120 | 1.50 |

**Table 1: Technical details of the UAV surveys.**

**3 Results**

**3.1 Electrical properties of the Gran Sometta rock glacier**

The 3D inverse model of data collected with a MG and DD configuration at 0.5 Hz is shown in Fig. 2. The inversion results
are presented in terms of the in-phase component $\sigma'$, the quadrature component $\sigma''$ and the phase $\varphi$ of the complex
conductivity, and visualized by slices cut through the 3D model vertically along the electrode lines. Along the longitudinal
profiles P8 (white lobe) and P5 (black lobe) we can identify three main layers: The conductive ($\sigma' > 10^{-4.25}$ S/m; or by its
inverse, the resistivity $\rho < 10^{4.25}$ Ωm) and polarizable ($\sigma'' > 10^{-6.5}$ S/m) top layer is related to unfrozen sediments and rocks
in the active layer with a thickness of approximately 4-6 m, in this case associated to blocky but also fine-grained materials
(Bearzot et al., 2022). No direct validation data of active layer thickness for the measurements in 2022 and 2023 is available
because boreholes with temperature sensors in the Gran Sometta rock glacier were destroyed in 2018. However, the active
layer thickness estimation is in a similar range as the value determined from the boreholes (6 m in borehole 1 and 4 m in
borehole 2) during 2015-2016 (see Fig. A2). We expect an active layer thickness in 2022 still in a range between 4 and 6 m



as rock glaciers are more resilient towards increasing air temperatures compared to other permafrost features (e.g., Pruessner et al., 2021). The second layer along the white and the black lobe is resistive ($\sigma' < 10^{-4.25}$ S/m, i.e., with a resistivity $\rho > 10^{4.25}$ $\Omega$m) and less polarizable ($\sigma'' < 10^{-6.5}$ S/m) corresponding to the frozen materials with a thickness of 20-30 m, in agreement with results from Bearzot et al. (2022). The bottom layer corresponds to the unfrozen materials related to higher
electrical conductivity and polarization values (i.e., $\sigma'$ and $\sigma''$).

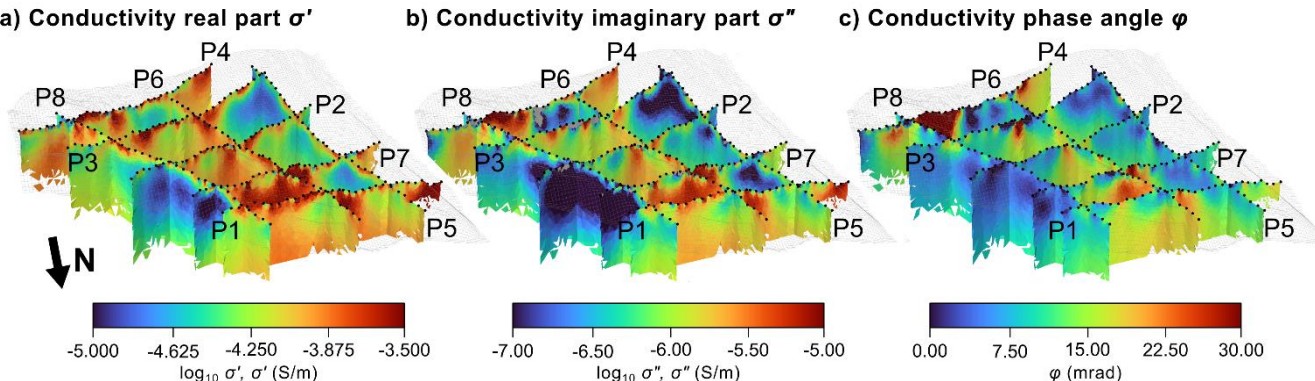

**Figure 2: Slices of the 3D inverse model along the profiles at 0.5 Hz in terms of a) real part, b) imaginary part and c) phase angle of the complex conductivity. The north direction is given by a black arrow. The positions of electrodes**
**are indicated by black dots.**

While the electrical properties along P5 and P8 are horizontally layered with a relatively continuous permafrost layer, we can observe higher spatial variability in the center of the rock glacier, at the position where both the white and the black lobe intersect. Such area is covered by all profiles apart from P5 and P8 and can be characterized by maximum values in $\sigma'$ and
$\sigma''$ in all layers. As discussed later, the higher $\sigma'$ and $\sigma''$ values in this area may be due to a lower and only sporadic ice content, the accumulation of finer materials, as well as due to water flowing internally along the rock glacier.

**3.2 Estimation of hydrogeological parameters based on IP data**

To estimate water content $\theta$ and hydraulic conductivity $K$ based on the DSLM in the Gran Sometta rock glacier, we calculated $M_n$ as the difference between the conductivity real part at 25 Hz and 0.1 Hz of the MG inverse model using
equation 8 proposed in Revil et al. (2020). However, this equation is only valid in the case of a constant phase angle and a linear relation between $M_n$ and $\sigma''$. In our case, the linear relation between $M_n$ and $\sigma''$ is weak (see Fig. 3c) with a large contrast between the predicted and the estimated slope $\alpha$ (10.11 and 3.37). Such discrepancy is due to the strong increase of $\varphi$ and the non-linear increase of $|\sigma^*|$ at frequencies above 2.5 Hz as presented in Fig. 3a (here marked as 'High frequency range'). Above 2.5 Hz, the surface conductivity is not only controlled by the accumulation and polarization of charges within
the EDL at the grain-fluid interface, but also at the ice-water interface as observed by Stillman et al. (2010). The contribution




of ice to the polarization at high frequencies is not considered in the DSLM (Coperey et al., 2019); thus, we decided to calculate $M_n$ based on the conductivity resolved at 0.1 and 2.5 Hz (in Fig. 3a marked as 'Low frequency range'). As presented in Fig. 3b, such analysis results in a stronger correlation between $M_n$ and $\sigma''$ with similar predicted and estimated values for $\alpha$ (2.06 and 1.91). However, the smaller the frequency window for the calculation of $M_n$ is, the higher is the

probability of overlooking polarization effects at higher frequencies, apart from ice polarization.

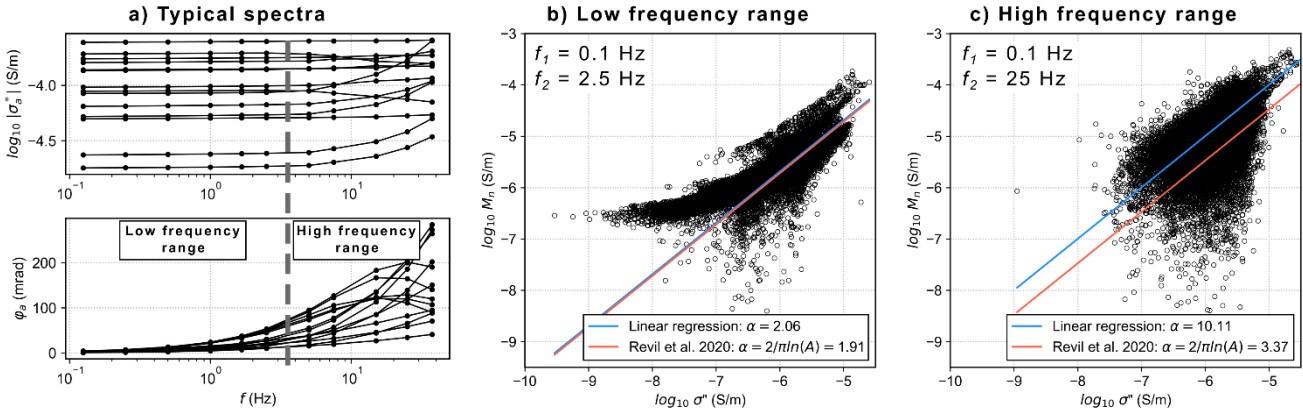

**Figure 3: a) Typical spectra of the apparent conductivity magnitude and phase shift of the Gran Sometta rock glacier. b) and c) present the relation between the normalized chargeability, calculated from a low frequency $f_1$ and a**

**high frequency $f_2$, and the imaginary part of the complex conductivity at the geometric mean frequency between $f_1$ and $f_2$, for the low frequency (b) and the high frequency range (c). The blue line indicates a linear regression, while the red line represents the theoretical relation (Revil et al., 2020). The data shown here were collected with a MG configuration along P1-P8.**

The results for $\theta$ and $K$ based on the DSLM are presented in Fig. 4 as slices parallel to the surface, in a depth of 4 and 20 m, as resolved from the 3D inversion. In the Appendix (Fig. A3) we added the same slices in terms of the conductivity at 0.1 Hz and the normalized chargeability. The water content is in a range of 0-15% (see Fig. 4a and 4b), and the hydraulic conductivity in a range of $10^{-7}$ to $10^{-3}$ m/s. Minimum values of both parameters ($\theta < 5\%$ and $K < 10^{-6}$ m/s) occur along P5 and P8 at a depth of 20 m, most likely because in this area most of the water is permanently frozen hindering water flow and

leading to a low $\sigma'$ and low $\sigma''$ at 0.5 Hz, as shown in Fig. 2. We can observe higher $\theta$ and $K$ values in the 4 m depth slice ($K$ up to $10^{-4.5}$ m/s and $\theta$ up to 10%), which is unfrozen; and thus, allows water flow. Maximum $\theta$ values (up to 15%) can be observed in the western part of the white lobe (center of the rock glacier), not only in the active layer close to the surface (4 m depth slice), where the rock glacier thaws during summer, but also at a depth of 20 m. Both $\sigma'$ and $\sigma''$ images (c.f., Fig. 2) indicate no continuously frozen layer in this area; thus, water can infiltrate from the uppermost layer in deeper areas and




accumulates. The high values in both $\sigma'$ and $\sigma''$ suggest the accumulation of fine grains in this area that reduces hydraulic conductivities and permits the water to be stored.

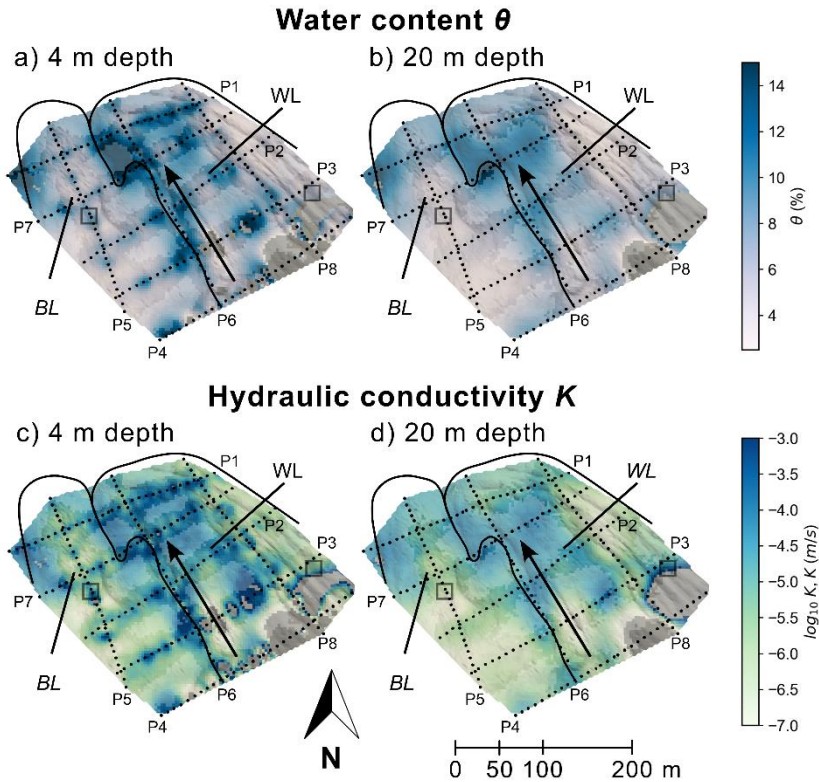

**Figure 4: Water content (a and b) and hydraulic conductivity (c and d) visualized as slices parallel to the surface in**
**two different depths. The edges of the black lobe (BL) and white lobe (WL) are indicated by black lines. Low sensitivity areas are displayed transparently, and the positions of electrodes are presented by black dots. Additionally, the direction of topography is indicated by a black arrow and the positions of tracer experiments by gray squares.**

### 3.3 Tracer experiments

Figure 5 presents the 3D inversion results of the small-scale tracer experiments at the white lobe (left) and the black lobe (right). The baseline (time step 0) electrical conductivity $\sigma'$ is visualized as slices along the two parallel electrode lines for the white lobe in Fig. 5a and for the black lobe in Fig. 5g. In the former, Fig. 5a reveals two layers, an upper conductive layer ($\sigma' > 10^{-4.25}$ S/m), which corresponds to the unfrozen active layer, and a lower continuous resistive layer ($\sigma' < 10^{-4.25}$ S/m), which is frozen and corresponds to the permafrost. The interface between both layers is at a depth of approximately 4
m (in agreement with borehole temperature data presented in Fig. A2). The $\sigma'$ baseline slices on the black lobe show the two



layers as well but we can observe that the contrast between the upper conductive and the lower resistive layer is weaker, and the depth of the interface varies along the parallel profiles. The comparison of Fig. 5a and 5g, reveals that the black lobe is related to slightly higher electrical conductivity values in the active layer, although we observed similar $\theta$ values in the active layer of the black and white lobe (see Fig. 4a). Hence, the high $\sigma'$ values resolved in the black lobe indicate a higher

content of fine grains filling large pores related to a significant contribution of surface conductivity.





**Figure 5: Slices presenting the real part of the complex conductivity along the electrode lines at the baseline for the white lobe (a) and black lobe (g). The black line indicates the interface between the active layer and the frozen layer, determined by the gradient of the electrical conductivity model. b) to f) and h) to l) represent the change of the conductivity with time after the injection of a salt tracer. Blue areas show a conductivity increase above 40% in the**





**case of the white lobe and above 20% in the case of the black lobe. The position of the tracer injection is indicated by a triangle, while the positions of electrodes are indicated by black dots. A scale was added on the bottom.**

The movement of the saltwater, injected in the center between the monitoring lines, can be resolved by means of the time-lapse changes in the ERT monitoring imaging results presented in Fig. 5b-5f (white lobe) and 5h-5l (black lobe). Blue colored voxels present a $\sigma'$ increase of $\geq 40\%$ (white lobe) and $\geq 20\%$ (black lobe) compared to the baseline $\sigma'$. In the white lobe, during and directly after the injection, the water propagates from the injection point downwards nearly perpendicular to the gravitational field. After reaching the interface between the active and the frozen layer, the saltwater starts to flow

through the active layer parallel to the interface. In the black lobe, the water percolates nearly vertically from the injection point, similar to the water movement in the white lobe directly after the injection. However, in the black lobe the saltwater plume stops at a distance of 2-3 m below the injection point and no flow downslope is observed; thus, suggesting a lower hydraulic connectivity in the black than in the white lobe (see discussion in section 4.1).

Changes in the electrical conductivity over time (after the injection of the tracer) permits to estimate the velocity of the water

moving through the subsurface, as presented in Fig. 6a. Singha and Gorelick (2005) and Camporese et al. (2011) demonstrated that the timing of the peak arrival is a reliable parameter for $K$ estimations because it is relatively unaffected of inversion artifacts. Hence, we used the time of the maximum conductivity concentration in 18 voxels to estimate the arrival time of the saltwater in the given voxels. Skipping the inversion and estimating $K$ based on raw data can lead to wrong interpretations, considering that the raw data does not contain information about the spatial position of the anomaly

(Camporese at al., 2011). As presented in Fig. 6a and 6c, the time $t$ of the maximum $\Delta\sigma$, also called arrival time $T_a$, increases with increasing distance from the injection position as the tracer moves from the injection position downslope through interconnected pores in the active layer. Additionally, we can observe that the slope of $T_a - \Delta\sigma$ is not constant over a distance of 0-23 m from the injection position but reveals a contrast between the nearest voxels and voxels which are further away from the injection position. This contrast reflects the two different flow characteristics identified in Fig. 5: An initial

nearly vertical flow (from now on called stage 1, S1) followed by a flow parallel to the interface between the active layer and the frozen layer (from now on called stage 2, S2).





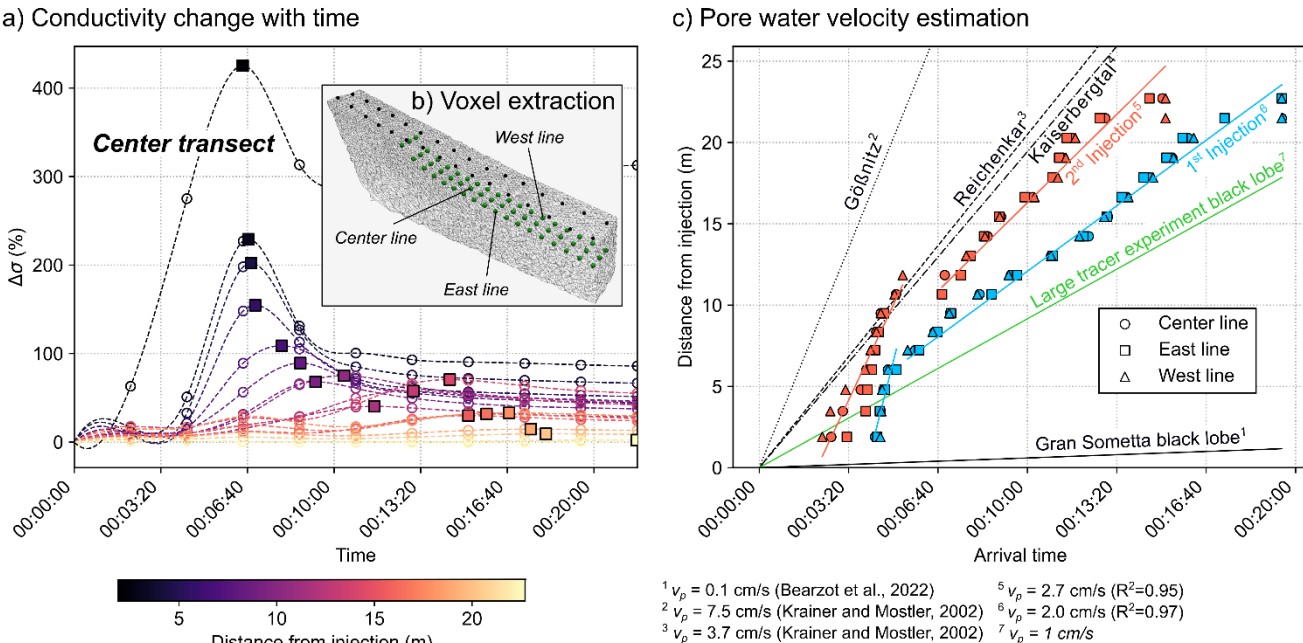

**Figure 6: a) Change of the conductivity in % with time relative to the baseline for different voxels (positions are presented in b)) along the center line of the tracer path in the active layer. The distance of the voxel from the injection position is presented by the line color, the maximum change of conductivity for each voxel is visualized by a square symbol. c) shows the time of the maximum change of conductivity for each voxel in relation to the distance from the injection point. The slope of a linear regression (in blue for injection 1, in red for injection 2) gives the pore water velocity. Pore water velocities of other studies are visualized by the black lines for a comparison. The estimated pore water velocity in the large-scale tracer experiment on the black lobe is added in green.**

The pore water velocity $v_p$, which was estimated by the slope between $T_a$ and the distance from the injection point, is presented in Fig. 6c. Due to the different slopes observed in Fig. 6a we fit two linear regression models per tracer injection, one for S1 and one for S2, resulting in a $v_p$ of 10.9 cm/s (injection 1) and 5.9 cm/s (injection 2) for S1 and a $v_p$ of 2.0 cm/s (injection 1) and 2.7 cm/s (injection 2) for S2. The contrast in $v_p$ between S1 and S2 is mainly related to the difference in the hydraulic gradient, while the discrepancy in $v_p$ between injection 1 and injection 2 is related to a difference in the matrix potential. When we injected saltwater for the second time the subsurface materials were already wet from the previous injection (higher $\theta$). The higher water content during the second injection leads to a lower absolute matrix potential and in turn to higher $K$ and $v_p$ values (e.g., Van Genuchten, 1980). Additional lines in Fig. 6c present results from different studies and will be discussed in section 4.2.



The TLBC inversion results of the large-scale tracer experiment on the black lobe are visualized in Fig. A4. In comparison to the small-scale tracer tests, the changes in the electrical conductivity close to the injection point and downslope were much smaller (~5% compared to ~20% and ~40%, respectively) most likely due to the different sensitivity associated with the larger electrode separation used in the large-scale tracer experiment (5 m vs. 2 m). In the images, the smaller changes in electrical conductivity associated with the tracer highlight conductivity changes associated to inversion artifacts, as can be observed particularly in the area above the injection point, where we do not expect big changes over time. Only interpreting the increase of electrical conductivity from the injection point along the surface downslope, the inversion images suggest a pore water velocity of around 1 cm/s (see green line in Fig. 6c). However, this value is not as reliable as the one derived from the small-scale tracer experiment on the white lobe due to time-lapse inversion artifacts.

### 3.4 Rock glacier movement

Figure 7 presents the direction and magnitude of the horizontal displacement of 161 rocks on the surface of the Gran Sometta rock glacier between August 2022 and August 2023 as determined from the kinematic analysis of the UAV images (see section 2.7). The kinematic changes of the rock glacier surface over the longer time span (2020-2023) are presented in Fig. A5 (Appendix). Most parts of the rock glacier moved to the northwest due to the aspect of the surface topography, except for a few points at the lateral boundaries of the rock glacier lobes. Over the whole area under investigation, the displacements are between 0 and 2 m per year, with a higher range of values on the black lobe (1-1.5 m/y) than on the white lobe (0-0.5 m/y). Maximum values can be observed on the southern part of the black lobe (> 1.5 m/y), while the minimum displacements (< 0.25 m/y) were measured on the front and eastern part of the white lobe.





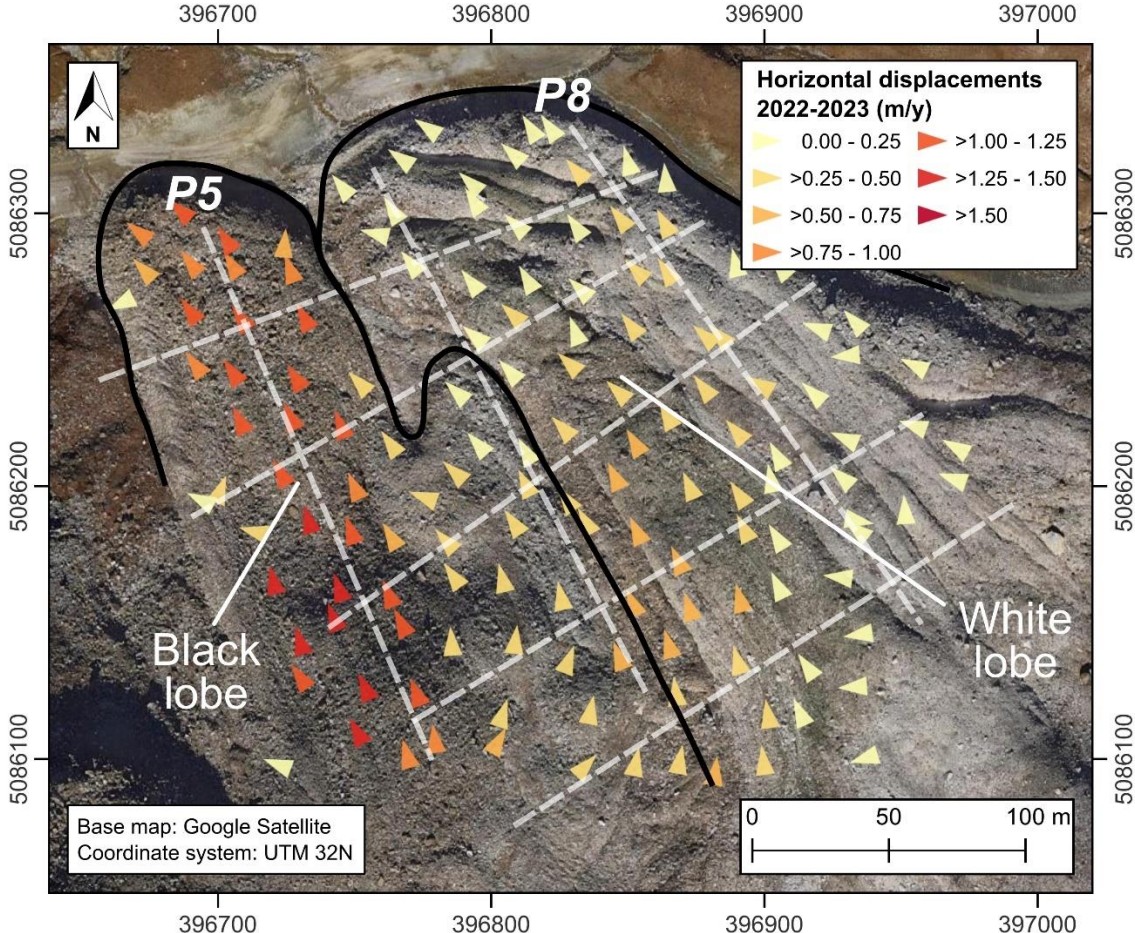

**Figure 7: Horizontal displacements of the Gran Sometta rock glacier between 2022 and 2023. The direction of the arrows presents the direction of the rock glacier movement, while the color displays the velocity of the rock glacier in m/yr. The positions of the SIP profile are indicated by white dashed lines, while the front edge of the rock glacier and the border between the black and the white lobe are presented by a black line. Orthophoto: © Google Earth 2022.**

## 4 Discussion

### 4.1 Identifying hydraulic units in the Gran Sometta rock glacier

In this study, we applied the model proposed by Soueid Ahmed et al. (2020) (explained in section 2.4, see Eq. (13)), based on the DSLM developed by Revil (2013a,b), to estimate $K$ on the Gran Sometta rock glacier. The results show a contrast in $K$ values between the unfrozen active layer ($K$ up to $10^{-3}$ m/s) and the frozen layer ($K$ down to $10^{-7}$ m/s). Such contrast in $K$ values is in agreement with results of the tracer test in the white lobe, where we injected saltwater on the surface and monitored associated spatio-temporal changes through 3D time-lapse electrical conductivity monitoring. The time-lapse



inversion images in the white lobe (c.f., Fig. 5) suggest that, after a nearly vertical flow in the uppermost 3 m, the injected water was blocked by a frozen layer. This layer might hinder the infiltration of most of the water into deeper areas along the tracer test site on the white lobe, which leads to a nearly parallel flow to the interface between the frozen layer and the active 510   layer.

On the black lobe, we conducted two tracer experiments with two different scales. In the small-scale experiment (c.f., Fig. 5), the injected water spread a few meters around the injection point but did not move further downslope. Compared to the tracer test area on the white lobe, we observed bigger blocks on the black lobe with huge pores and a poor hydraulic connection. We suggest that we did not inject enough water (70 l per injection) to hydraulically connect such big pores; thus, 515   the water might have been run through the uppermost hollow spaces, split up in different holes and stopped there. Additionally, the conditions during the tracer experiment in the black lobe in August 2023 were drier than during the tracer test in the white lobe in October 2022, which also leads to a lower hydraulic connection between pores. The saltwater injected in the large-scale tracer experiment in the black lobe has been successfully tracked in the inversion images (c.f., Fig. A4) but the results need to be considered carefully due to time-lapse inversion artifacts observed along the profile. The 520   images reveal a similar water flow as observed in the small-scale tracer experiment in the white lobe with a nearly parallel flow to the surface through the active layer.

The water flow resolved with electrical images described above is in agreement with the assumption that frozen areas in active rock glaciers act as impermeable layers hindering water flow (e.g., Krainer and Mostler, 2002; Giardino et al., 2011; Winkler et al., 2016 and Del Siro et al., 2023). Krainer and Mostler (2002) e.g., conducted hydrological tracer experiments 525   on rock glaciers in Austria, where the time between the tracer injection and the arrival time at the rock glacier spring allows to measure a mean pore water velocity, which resulted in relatively large flow velocities. Given that frozen materials are rather poor hydraulic conductors (e.g., Burt and Williams, 1976) and the active layer mainly consists of large blocks with huge pores with a high hydraulic conductivity, the authors suggested that the water does not percolate into the frozen layer but flows through the active layer. However, such tracer tests do not provide information about the actual flow path of the 530   tracer. Recently, Pavoni et al. (2023b) conducted a tracer experiment in the inactive Sadole rock glacier, where they tracked the position of a salt tracer by a 2D ERT monitoring over a distance of 40 m. They reported that most of the water did not percolate into the frozen area but moved through the active layer.

Bearzot et al. (2022) injected a tracer fluid in the black lobe of the Gran Sometta rock glacier and measured the arrival time at the spring located at a distance of around 270 m down gradient. The arrival of the tracer was identified by changes in the 535   chemical composition of the spring water, and estimations of the pore water velocity were conducted through the distance between the injection point and the spring position divided by the elapsed time. Their analyses in the black lobe resolved a pore water velocity of around $10^{-3}$ m/s but they only monitored the chemical composition of the spring water during the night, which might have led to an underestimation of the pore water velocity. The value is smaller than the pore water velocities estimated by Krainer and Mostler (2002) ($v_p \sim 10^{-2}$) for three rock glaciers in Austria (see Fig. 6), but still suggests 540   that the water mainly flew through the active layer.



## 4.2 Hydraulic conductivity estimation based on tracer experiments and multi-frequency electrical conductivity

Based on the arrival times of the tracer and the time elapsed, we estimated the pore water velocity $v_p$ in the white lobe for the two stages (c.f., Fig. 6c), resulting in values of 8.4 cm/s (S1) and 2.35 cm/s (S2) respectively (mean value over injection 1 and injection 2). Multiplying $v_p$ with the reciprocal of the hydraulic gradient (0.57 for S1 and 4.77 for S2) and an assumed

mean porosity of 40% (as used in other rock glacier studies, e.g., Hauck et al., 2011; Halla et al., 2021) allows us to estimate the hydraulic conductivity, which results in $K$ values of $1.9 \times 10^{-2}$ m/s (S1) and $4.5 \times 10^{-2}$ m/s (S2). The difference in the values resolved for the two stages is related to vertical changes in the water content in the active layer of the rock glacier. While the uppermost rocks in the active layer (S1) are relatively dry, the material at the interface between the frozen layer and the active layer (S2) can assumed to be more humid leading to the slight increase of $K$ in S2.

The large-scale tracer experiment on the black lobe permitted the computation of the pore water velocity (see Fig. A4 in the Appendix), resolving for values around 1 cm/s (see green line in Fig. 6c), which is half of the $v_p$ estimated for the white lobe. This lower hydraulic conductivity found in the black lobe compared to the white lobe would explain the difference in water movement observed in the small-scale tracer experiments conducted on the white and the black lobe. However, the $v_p$ value estimated by Bearzot et al. (2022) for the black lobe (based on the analysis of water samples at the rock glacier spring

after the injection of a tracer fluid) is one order of magnitude lower than the value derived from our tracer experiment ($10^{-3}$ m/s vs. $10^{-2}$ m/s). This discrepancy can be related to the difference in the covered flow path. While our ERT monitoring covered a steep area with a maximum length of only 30 m, Bearzot et al. (2022) investigated the flow over a distance of 270 m, covering also flat areas. Accordingly, the study area of Bearzot et al. (2022) might result in the averaged values from high pore water velocity in the steep areas (covered in our study) and the low pore water velocity in the flatter areas not

investigated with our tracer test.

In Fig. 6c, we also compare $v_p$ estimations in our study with results from other rock glaciers based on hydrological tracer experiments. The $v_p$ results for the Gößnitz, Reichenkar and Kaiserbergtal rock glaciers (located in the Eastern Alps) reveal $v_p$ values varying between 3.5 to $7.5 \times 10^{-2}$ (Krainer and Mostler, 2002). Those values are in the same order of magnitude as the value estimated for the white lobe with the small-scale tracer experiment and the value estimated in the large-scale tracer

experiment in the black lobe. Variations in $v_p$ between different rock glaciers can be related to different slope angles, geologies, active layer depths and ice content.

Comparing the $K$ values from the tracer experiment in the white lobe with the results of $K$ estimations for the active layer based on the analysis of the multi-frequency conductivity data (c.f., Fig. 4c) reveals a discrepancy of two magnitudes ($10^{-2}$ m/s vs. $10^{-4}$ m/s). Such a discrepancy might be most likely related to the enhanced hydraulic connectivity associated with the

addition of water in the tracer experiments. Increasing the saturation enhances the interconnection between pores (e.g., Western et al., 2001); thus, resulting in higher $K$ values. Moreover, the approach based on the analysis of multi-frequency conductivity data assumes a constant value in the fluid conductivity ($\sigma_f = 0.01$ S/m), the saturation exponent ($n = 2$) and cementation exponent ($m = 2$) across the entire rock glacier, as described by Soueid Ahmed et al. (2020) and Revil et al.





(2020), while the analysis of the tracer test does not require such assumptions. Accordingly, while our study reveals that the use of the DSLM might provide relevant information about the hydrogeological features and their connectivity, a quantitative estimation of water content and hydraulic conductivity might still require some calibration for its application. Additionally, the variations in the estimated values might be related to the electrode spacing (7.5 m) used for the SIP mapping, which reduces the resolving capability of the electrical properties in the uppermost layer. An electrode spacing of 7.5 m was chosen as a compromise between a large depth of investigation (needed to cover the area below the permafrost), and a high resolution to resolve the active layer thickness, as well as to collect data over a large part of the rock glacier in a reduced time. Future studies might consider the collection of data with varying electrode spacing to improve hydrogeophysical investigations.

### 4.3 The large variability in water content and hydraulic conductivity at the Gran Sometta rock glacier

Results of the estimation of hydrogeological properties ($\theta$ and $K$) have shown a high spatial variability with depth and laterally across the rock glacier (c.f., Fig. 4). We can observe high $\theta$ values (up to 15%) in the active layer, where rain and snow meltwater can accumulate on top of the frozen layer. At a depth of 20 m, the rock glacier is frozen in several areas resulting in a low amount of liquid water, and which hinders vertical water flow to deeper areas. Laterally, local maxima of $\theta$ can be found along furrows, particularly longitudinal furrows, while local minima can be found in ridges. Such spatial pattern suggests that water from snowmelt or rainfalls flows nearly vertically through the active layer, as described by S1 in the small-scale tracer experiments, until it reaches the interface between the active layer and the frozen layer. After that, in S2 the water moves along the interface downhill in direction of the steepest slope, either in flow direction of the rock glacier or into a near depression zone (see e.g., Halla et al., 2021), which are the most shaded areas on the rock glacier with low evaporation rates.

The longitudinal furrows act as preferential flow paths transferring water from areas above to the front part of the rock glacier (Halla et al., 2021) as the furrows are related to high hydraulic conductivity ($K > 10^{-4.5}$) (see Fig. 4c). After flowing through the furrows, water mainly accumulates in the western front part of the white lobe and can percolate into deeper areas (20 m depth) because there is no continuously frozen horizontal layer in such area. The complex conductivity results have revealed that the internal structure of the western frontal part of the white lobe is different compared to the eastern part of the white lobe and the black lobe. There is no continuous layer of low conductivity, associated with frozen materials, which might be related to a high ice-degradation rate allowing water to infiltrate into deeper areas, where we observed high $\theta$ values at a depth of 20 m, and which in turn accelerates the degradation process (Ikeda et al., 2008).

### 4.4 Spatial variability of rock glacier movement is dominated by water content in the shear horizon

Several studies working on the kinematics of rock glaciers have revealed an overall acceleration of rock glacier movement during the last decades (e.g., Delaloye et al., 2010; Marcer et al., 2021; Kellerer-Pirklbauer et al., 2024). In some studies, variations in rock glacier velocities were related to the topography (e.g., Müller et al., 2016; Bodin et al., 2018), to the ice





content (e.g., Kääb et al., 2007) or the thermal state (e.g., Bearzot et al., 2022) because, e.g., topographical variations and ground surface temperatures can be measured directly. Recent studies discuss the role of water content (e.g., Kenner et al., 2019), with a particular focus on the amount of liquid water in the shear horizon (e.g., Cicoira et al., 2019), where most of the deformation takes place (Arenson et al., 2002). However, due to the limited access, direct measurements of the water

content in the shear horizon are challenging. Phillips et al. (2023) installed piezometers and an ERT monitoring in boreholes on a rock glacier in Switzerland that penetrate until the upper boundary of the shear horizon, permitting a direct investigation of the shear plane. They reported a link between high water content and increased rock glacier velocity for two seasons (Bast et al., 2024).

Kinematic analyses on the Gran Sometta rock glacier have shown a contrast in surface velocities between the white and the

black lobe, with higher values on the black lobe (~1-1.5 m/y) than on the white lobe (0-0.5 m/y) (c.f., Fig. 7 and A4). Such contrast is in agreement with kinematic results of similar analyses published by Bearzot et al. (2022) for the period 2016-2019 (~1.2 m/y for the black lobe and ~0.4 m/y for the white lobe). Bearzot et al. (2022) related the spatial flow patterns to the internal structure and the contrast in topography with a larger slope angle on the black lobe than on the white lobe. The geophysically based estimation of the hydrogeological properties along both lobes in this study (see Fig. 8) allows to

investigate the relation between the water content and the spatial variations in rock glacier velocity.

Figure 8c,g show that in the frozen layer most of the water is frozen in both lobes ($\theta < 5\%$) leading to a low electrical conductivity (see Fig. 8a). Below this low electrical conductivity layer, we observe a layer of increased electrical conductivities associated with increased water content. In this area, higher $\theta$ is found for the black lobe (up to 10%) than for the white lobe (around 5%). Our results might indicate that water accumulates in the shear horizon of the black lobe at a

profile distance above 100 m in southern direction, with higher $K$ values ($> 10^{-5}$ m/s) above 100 m profile distance than downslope ($< 10^{-6}$ m/s) (see Fig. 8d,h). The lower hydraulic conductivity values may be due to a larger content of fine-grained sediments, as also suggested by the higher electrical conductivity and normalized chargeability values (Fig. 8a,b). Such low $K$ area might reduce the velocity of water flow coming from uphill leading to an increase in pore water pressure, which would in turn increase the potential of deformation in the shear horizon and can further accelerate the rock glacier

movement. This observation is similar to the one reported by Kenner et al. (2017) at the Ritigraben rock glacier, where borehole data revealed a wet area below the frozen layer at the depth of the shear horizon. Kenner et al. (2019) argue that the idea of water content as driving force for rock glacier kinematics is not a contradiction to the correlations between temperature and rock glacier velocity variations temperatures (Kääb et al., 2007; Roer et al., 2008), rather water content can act as the link between temperature and kinematic changes.



**Figure 8: a) and e) show the real part of the complex conductivity, b) and f) the normalized chargeability, c) and g) the water content and d) and h) the hydraulic conductivity along P5 (left) and P8 (right). Pixels where the normalized chargeability results in negative values are blanked. The positions of electrodes are indicated by black dots, the interface between the active layer and the frozen layer is presented by a gray line. The rock glacier flow direction of each lobe is presented by a black arrow including its mean velocity.**





## 5 Conclusions

In this study, we investigated the hydrogeological properties of the active Gran Sometta rock glacier in the Italian Alps by an SIP mapping. Electrical conductivity and normalized chargeability data were used within the DSL model to directly estimate

the spatial water content and hydraulic conductivity distribution within the rock glacier. To validate the estimation of both parameters, we conducted tracer experiments, where we injected saltwater and tracked the change of electrical conductivity in the subsurface by an ERT monitoring. Both methods reveal similar hydraulic units: an upper non-frozen layer with high hydraulic conductivity and a lower frozen layer with low hydraulic conductivity. However, in the absolute values there is a discrepancy of two magnitudes between the hydraulic conductivity estimated within the DSL model and the tracer

experiment, which is mainly related to the decreased resolution of the SIP mapping in the uppermost layer due to a large electrode spacing.

The Gran Sometta rock glacier can be divided in three different areas characterized by their hydrogeological properties. As revealed by complex conductivity images, the internal structure of the black lobe and the eastern part of the white lobe is similar with an active layer of around 4-6 m thickness and a continuously frozen layer beneath, which is in agreement with

borehole temperature data. The hydraulic conductivity and water content estimations based on the DSLM have shown that in both areas the frozen layer acts as a hydraulic barrier hindering water flow into deeper areas. Most of the water coming from rainfall or snowmelt flows rapidly through the active layer with hydraulic conductivities in a range of $10^{-2}$ m/s, with slightly higher values in the white lobe than in the black lobe, as revealed by the tracer tests. Contrary to this, in the western front part of the white lobe, ice content is low, and a continuously frozen horizontal layer is missing leading to water penetration

into deeper areas where water accumulates in fine-grained sediments.

Additional analyses of the hydrogeological properties in the area below the poorly electrically conductive ice-rich layer have shown a higher water content in the black lobe (up to 10%) than in the white lobe (around 5%). As kinematic analyses have revealed that the black lobe moves three times faster than the white lobe, such contrast in water content suggests that water accumulated close to the shear horizon decreases the frictional resistance and in turn might accelerate the rock glacier

velocity.

Our study shows that the spectral induced polarization method is a valuable tool to describe and quantify hydraulic conductivity and water content in rock glaciers. Such information does not only help in the understanding of the hydrological cycle in rock-glaciated alpine catchments, but it should also be considered in the analysis of spatial patterns in rock glacier movement.



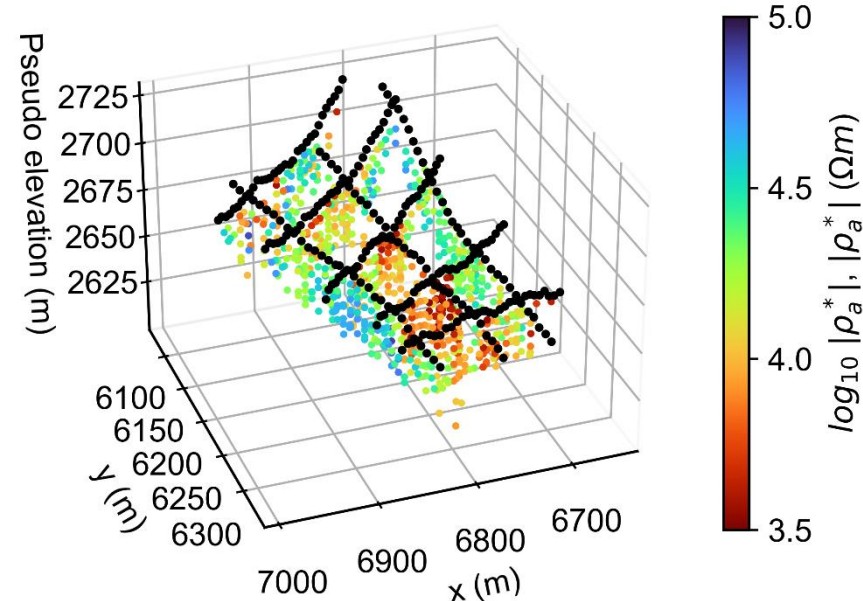

**Figure A1: 3D visualization of the data along P1-P8 after the filtering procedure. The data are presented in terms of the apparent resistivity ($|\rho_a^*|$). Electrode positions are visualized as black dots.**

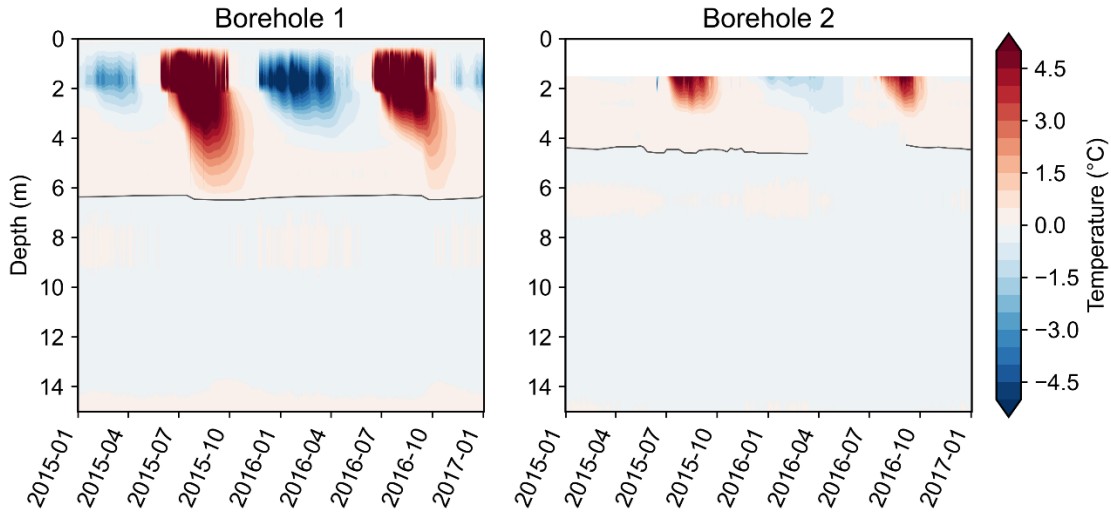


**Figure A2: Temperature data at two boreholes (borehole BH1 and borehole BH2) on the Gran Sometta rock glacier. The active layer depth is displayed by a gray line.**



**Borehole temperatures**

Temperature data from two boreholes in the Gran Sometta rock glacier are presented in Fig. A2. The active layer depth
(ALD), which is the depth until the material thaws at least once per year, is ~6 m and ~4 m in borehole 1 and 2, respectively.
The temperature data in borehole 1 shows that between 4 and 6 m depth the material does not freeze during winter, which
suggests the existence of a supra-permafrost talik. Taliks have been observed several times in the European Alps (e.g.,
Zenklusen Mutter and Phillips, 2012; Luethi et al., 2017) and are assumed to be an indicator of permafrost degradation, but
they have not been investigated in detail, yet. However, such information has not been reported at the Gran Sometta rock
glacier. Nonetheless, as the temperature in the talik is close to the freezing point, we cannot exclude any calibration error in
the temperature sensors, which could also lead to such temperature pattern.

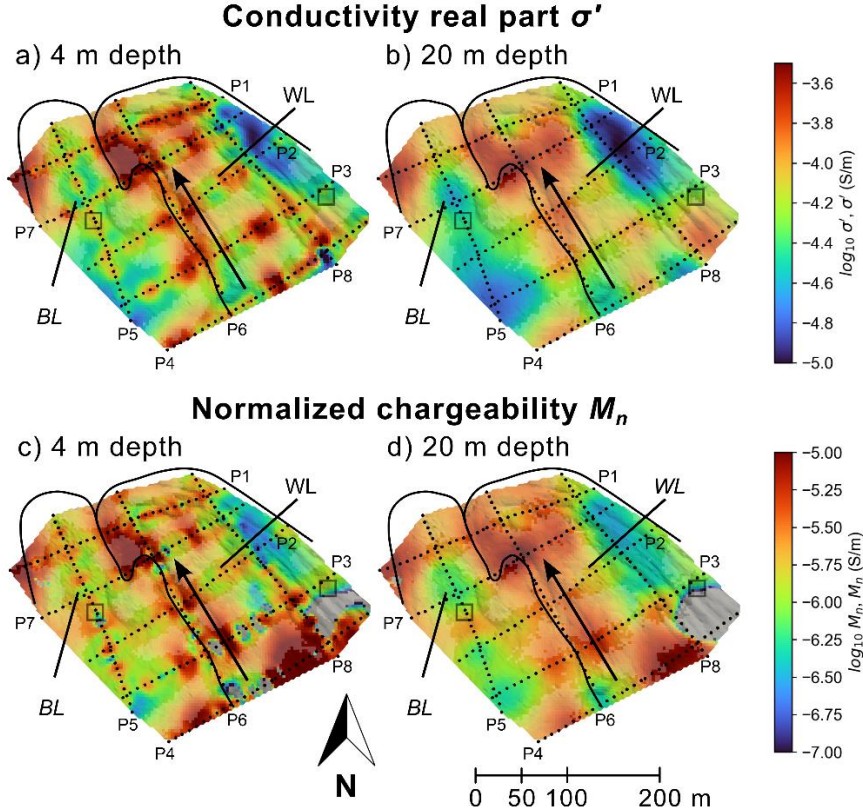

**Figure A3: Conductivity real part (a and b) and normalized chargeability (c and d) visualized as slices parallel to the
surface in two different depths. The edges of the black lobe (BL) and white lobe (WL) are indicated by black lines.
Low sensitivity areas are displayed transparently, and the positions of electrodes are presented by black dots.
Additionally, the direction of topography is indicated by a black arrow and the positions of tracer experiments by
gray squares. The normalized chargeability refers to the difference between the bulk electrical conductivity at high
and low frequency (2.5 and 0.5 Hz in our study, respectively), as presented in Equation 8.**



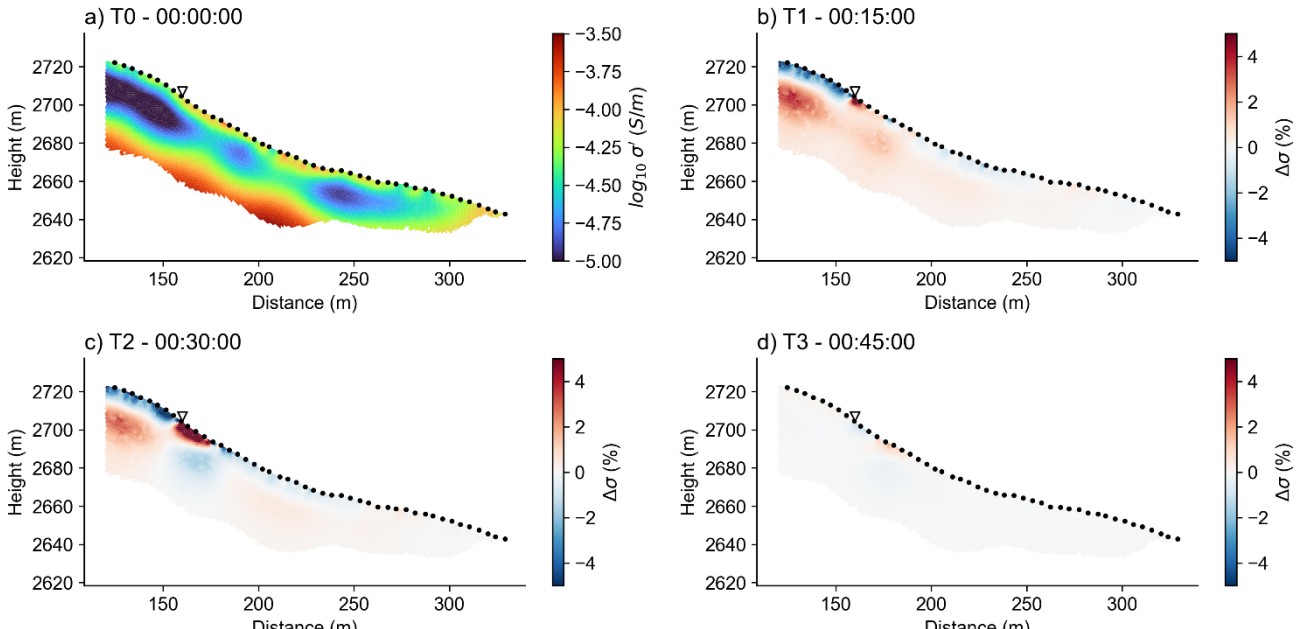

**Figure A4: 2D conductivity inversion results of the baseline data for the large-scale tracer profile in the black lobe (a). Subplots b) to d) present the change of conductivity relative to the previous time step in % (linear scale). The**

**position of the tracer injection, at the surface, is indicated by a triangle, while the positions of electrodes are indicated by black dots.**



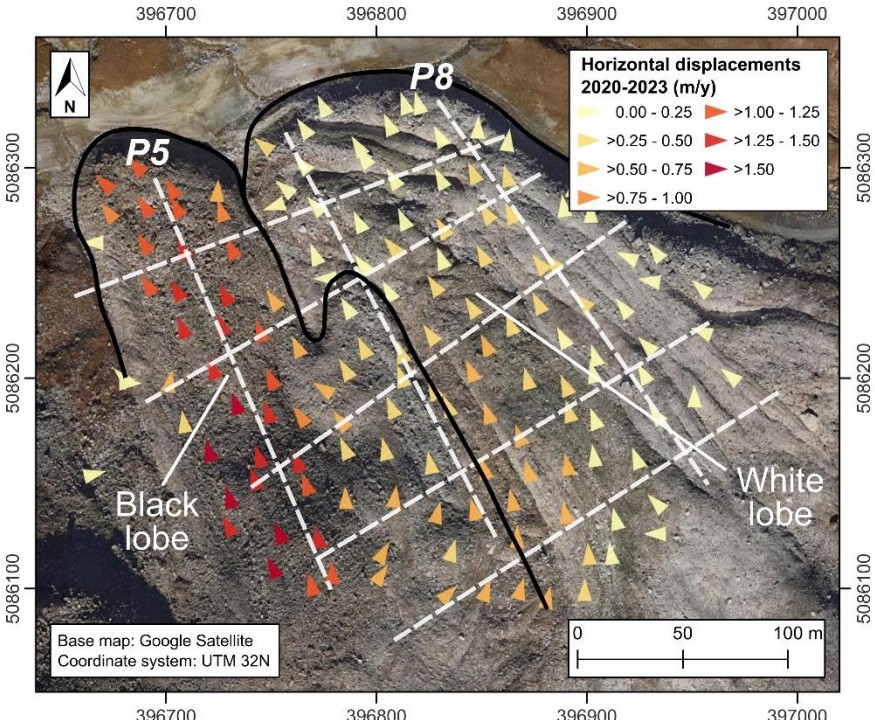

**Figure A5: Horizontal displacements of the Gran Sometta rock glacier between 2020 and 2023. The direction of the**
**arrows presents the direction of the rock glacier movement, while the color displays the velocity of the rock glacier in**
**m/yr. The positions of the SIP profile are indicated by white dashed lines, while the front edge of the rock glacier and**
**the border between the black and the white lobe are presented by a black line. Orthophoto: © Google Earth 2022.**

**Data availability**

The geophysical and kinematic data that support the findings of this study are available from the corresponding author upon
request.

**Author contributions**

AFO and CM designed the experimental setup, all authors planned and coordinated the field logistics. CM and CH collected
the geophysical data. CM processed the geophysical data, UMDC collected and processed the kinematic data. AFO, CH and
CM interpreted the geophysical results, and all authors discussed the results. CM led the preparation of the draft, where all
authors contributed actively, with major help from AFO and CH.



**Competing interests**

Some authors are members of the editorial board of The Cryosphere.

**Acknowledgements**

The authors are grateful to Sophia Keller, David Radlbauer, Elia Cornali and Ida Bürgermeister for their help in the
collection of the geophysical data, to Federico Grosso and Michel Isabellon for their logistical support and help in the
collection of the geophysical data during the field surveys, and to Theresa Maierhofer for the constructive comments on the
manuscript. We are thankful to the cable car company Cervino S.p.A for the logistical support and to ARPA VdA for
providing borehole temperature data. CM is thankful to ARPA VdA for hosting an Erasmus+ internship and to Erasmus+ for
the financial support.

**Financial support**

This research is part of the project Tipping Points and Resilience of Mountain Permafrost under Increasing Frequency of
Heat Waves (TREAT), which is financially supported by the Austrian Science Fund (FWF, grant no. I 6549-N).

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
