# Peer review of "Spectral induced polarization survey for the estimation of hydrogeological parameters in an active rock glacier"

_EGUsphere, 2024_

## Referee Comment (RC2)

[referee-annotated manuscript omitted]

---

## Author Comment (AC1)

**Reviewer #1**

| Reviewer Comment | Author's response |
|---|---|
| *GENERAL* | |
| 1) This study presents the results of using Spectral Induced Polarization (SIP) surveys for estimating hydrogeological parameters in an active rock glacier. In addition to SIP, the authors use tracer experiments and photogrammetry to compare various findings regarding rock glacier hydrology and movement. I believe this study is of strong interest to the community and provides an excellent field study to guide future improvements in understanding hydrological processes in such environments. The paper is very well organized and written, and I enjoyed reading it. | We thank the reviewer for the positive comment. |
| 2) My only main concern is regarding the discussion on the estimation of hydraulic conductivity. While the electrical approach seems to provide interesting results for delineating features and tracking the plume during the tracer experiment, I believe the discussion on how SIP is used to estimate hydraulic conductivity (K) lacks depth. I would appreciate it if the authors included more discussion on the various assumptions and sources of uncertainty. I suggest improving the discussion and possibly adding some rough uncertainty bounds to the various parameters used to parameterize the equations and evaluating the impact of those (e.g., using ko on line 233) on the estimated values. | We agree with the reviewer's comment. The main challenge of giving an uncertainty to the parameters estimated (e.g., water content ($\theta$) and hydraulic conductivity ($K$)) is the uncertainty of the inverse model parameters (complex conductivity) because of its non-uniqueness. Estimating the uncertainty of the geophysical parameters would require extensive analyses and would be beyond the scope of this manuscript. In the manuscript we will try to estimate the uncertainties of the parameters (e.g., fluid conductivity) used in the equation which describes $\theta$ and $K$ and use Gauss's propagation of error to estimate the uncertainty of the final parameters ($\theta$ and $K$). However, we always need to keep in mind that the uncertainty of $\theta$ and $K$ might be biased by the uncertainty of the inverse model parameters. |
| | |
| *Specific* | |
| 1) L.65: "not suited". To my knowledge, drilling boreholes for monitoring temperature is still the most reliable approach to monitor thermal dynamics. Consider rephrasing this sentence. | We thank the reviewer for the comment. We agree that this sentence was not well formulated, and boreholes are still the most reliable approach to monitor thermal dynamics. The term "not suited for monitoring applications" was related to ice content estimation. When drilling a borehole, the ice content in the samples can be quantified but only once at the time of the drilling. We will reformulate the sentence. |
| 2) L.117: Including a philosophical tone in Latin is a nice touch, but it may be unclear for the reader… and accidental. | We thank the reviewer for the comment. The sentence was accidentally added when |

| | |
|---|---|
| | converting to the template of The Cryosphere. We will remove it. |
| 3) L.185: Consider splitting this sentence into two. | We will split the sentence in two. |
| 4) L.189: "improves" is vague. I suggest "showed some promise to estimate..." and adding more details under specific conditions. | We agree with the comment of the reviewer. As the reviewer proposed, we will change the wording and add more details about the materials investigated in the cited studies. |
| 5) L.225 to 242: There are many values used from the literature to parameterize these equations. A more thorough discussion of these choices would be appreciated. For example, on line 223, more information on the ko value and the material it was defined for would be appreciated. ko likely has a significant impact on the k and K value and is a considerable source of uncertainty. Discussing these limitations more thoroughly, either here or in the discussion section, would be fair. Another point is the hydraulic gradient, which assumes the groundwater hydraulic gradient but is taken as the topography gradient here. Same for the porosity (0.4, on L. 238) and the implication on the estimates. | We agree with the reviewer's comment, which partly overlaps with general comment #2. We will add one paragraph in the discussion section where we discuss the uncertainty of the parameters (including e.g., hydraulic gradient and porosity) used to estimate $\theta$ and $K$ and where we use Gauss's propagation of error to estimate the uncertainty of the parameters of interest ($\theta$ and $K$). |
| 6) Figure 3: Consider showing the relationship between chargeability and the real part of conductivity (supplementary material would be fine). | The DSLM uses the normalized chargeability ($M_n$) to account for surface conductivity in the estimation of water content. In this study, we used IP measurements in the frequency-domain; thus, the parameter we get is the complex conductivity real ($\sigma'$) and imaginary part ($\sigma''$). To calculate $M_n$ we use the linear relationship between $\sigma''$ and $M_n$ proposed by Revil et al. (2017). Fig. 3 is used to evaluate whether this linear relationship is also valid for the data from the Gran Sometta rock glacier and which range of frequencies can be used. A plot with the relation (or discrepancy) between $\sigma'$ and $M_n$ would only show the influence of the electrolytic conductivity on the total electrical conductivity. Such relation is not directly used in this study, and we think that such plot might confuse the reader. |
| 7) L.437: Please provide an explanation for why you used >20% for the black lobe, or consider stating the maximum increase in white vs. black lobe. | In the black lobe the maximum change in conductivity was lower than in the white lobe. So, we used 20% to make the changes visible in the Figure. We will add an explanation in the text about the different threshold values. |
| 8) Figure 6 and associated text: It is unclear if the discussion is about the white lobe only or both lobes. Please clarify in the figure caption and associated text. | We agree with the reviewer's comment and will clarify that in the figure caption and the associated text. |
| 9) L.626: Please clarify your thoughts on the order of importance in parameter impact. The terrain slope (please provide an estimate and | We agree with the reviewer's comment. The topography might be an additional controlling parameter in the Gran Sometta rock glacier |

| discuss) seems much steeper in the black lobe and would be expected to be a strong control. Does that not complicate and maybe compromise the interpretation of another control using the current data set ? Please consider adding some discussion of the impact of the slope gradient alone. | velocity, as suggested by Bearzot et al. (2022) (see Line 618). We will highlight the role of the slope in the last paragraph. |
| --- | --- |

---

## Author Comment (AC2)

**Reviewer #2**

| Reviewer Comment | Author's response |
|---|---|
| *GENERAL* | |
| 1) The manuscript "Spectral Induced Polarization survey for the estimation of hydrogeological parameters in an active rock glacier" by Moser et al. describes a very interesting experiment on an active rock glacier. The authors used a suite of electrical measurements (SIP, ERT) to study the structure and hydrological properties of the glacier. The manuscript makes a good argument for the use of geophysical methods in glacier studies and should be published after the authors address the comments discussed below. | We thank the reviewer for the positive comment. |
| 2) First, the good news. The results are really encouraging for the use of electrical methods in such studies. The discussion is mostly well written, adequately discuss the data and the interpretation is very logical, with the proper caution that geophysical methods deserve.. This section really highlights the benefits of such methods for glacier studies. | We thank the reviewer for the positive comment. |
| 3) My main concert is not with the results and the discussion, rather with the presentation and the flow of the manuscript, For example the terminology used is inconsistent and can be confusing, especially to the non expert; acronyms / full terms are used interchangeably, while multiple names are used for the same term (can be introduced, but should use one for consistency). | We agree with the reviewer's comment and will improve the flow of the manuscript (see detailed answers on the reviewer's comments in the pdf document). |
| 4) The authors provide a lot of information, and use multiple results (and past studies), while this is good and needed, results in a manuscript with flow issues. I find the introduction confusing and difficult to follow; the authors introduce new terms, without fully addressing them and discuss some of the methods in not needed detail; I would suggest the authors use the introduction to just introduce the problem (as they do), discuss the need of a solution, and the approach and then introduce the methods only at high level; the details of the methods, including past studies, can go to a 'theory' part of the methods. Same for the results section - results are a mix of theory, results and interpretation making difficult to read / follow. | We will remove parts of the introduction (see detailed answers on the reviewer's comments in the pdf document). We also note here that some information about the geophysical method used in this study is included within the introduction considering that this technique might be less known within the audience of The Cryosphere. In the results section we will shift parts of it to the discussion to avoid mixing results and interpretation as proposed by the reviewer (see detailed answers on the reviewer's comments in the pdf document). |

[revised manuscript text omitted]